# Axonal Regrowth of Olfactory Sensory Neurons In Vitro

**DOI:** 10.3390/ijms241612863

**Published:** 2023-08-16

**Authors:** Rebecca Sipione, Nicolas Liaudet, Francis Rousset, Basile N. Landis, Julien Wen Hsieh, Pascal Senn

**Affiliations:** 1The Inner Ear and Olfaction Lab, Department of Pathology and Immunology, Faculty of Medicine, University of Geneva, Rue Michel Servet 1, CH-1211 Geneva, Switzerland; rebecca.sipione@unige.ch (R.S.); francis.rousset@unige.ch (F.R.); basile.landis@hcuge.ch (B.N.L.); pascal.senn@hcuge.ch (P.S.); 2Bioimaging Core Facility, Faculty of Medicine, University of Geneva, Rue Michel Servet 1, CH-1211 Geneva, Switzerland; 3Rhinology-Olfactology Unit, Department of Otorhinolaryngology—Head and Neck Surgery, Geneva University Hospitals, 4 Rue Gabrielle-Perret-Gentil, CH-1211 Geneva, Switzerland

**Keywords:** anosmia, traumatic brain injury, axonal regeneration, axon tracing, nerve growth factors, olfactory neurons

## Abstract

One of the most prevalent causes of olfactory loss includes traumatic brain injury with subsequent shearing of olfactory axons at the level of the cribriform plate (anterior skull base). Scar tissue at this level may prevent axonal regrowth toward the olfactory bulb. Currently, there is no cure for this debilitating and often permanent condition. One promising therapeutic concept is to implant a synthetic scaffold with growth factors through the cribriform plate/scar tissue to induce neuroregeneration. The first step toward this goal is to investigate the optimum conditions (growth factors, extracellular matrix proteins) to boost this regeneration. However, the lack of a specifically tailored in vitro model and an automated procedure for quantifying axonal length limits our ability to address this issue. The aim of this study is to create an automated quantification tool to measure axonal length and to determine the ideal growth factors and extracellular proteins to enhance axonal regrowth of olfactory sensory neurons in a mouse organotypic 2D model. We harvested olfactory epithelium (OE) of C57BL/6 mice and cultured them during 15 days on coverslips coated with various extracellular matrix proteins (Fibronectin, Collagen IV, Laminin, none) and different growth factors: fibroblast growth factor 2 (FGF2), brain-derived neurotrophic factor (BDNF), glial cell-derived neurotrophic factor (GDNF), nerve growth factor (NGF), retinoic acid (RA), transforming growth factor β (TGFβ), and none. We measured the attachment rate on coverslips, the presence of cellular and axonal outgrowth, and finally, the total axonal length with a newly developed automated high-throughput quantification tool. Whereas the coatings did not influence attachment and neuronal outgrowth rates, the total axonal length was enhanced on fibronectin and collagen IV (*p* = 0.001). The optimum growth factor supplementation media to culture OE compared to the control condition were as follows: FGF2 alone and FGF2 from day 0 to 7 followed by FGF2 in combination with NGF from day 7 to 15 (*p* < 0.0001). The automated quantification tool to measure axonal length outperformed the standard Neuron J application by reducing the average analysis time from 22 to 3 min per specimen. In conclusion, robust regeneration of murine olfactory neurons in vitro can be induced, controlled, and efficiently measured using an automated quantification tool. These results will help advance the therapeutic concept closer toward preclinical studies.

## 1. Introduction

Olfactory dysfunction (OD) manifests itself primarily in the partial (hyposmia) or total (anosmia) inability to smell odors, including the diminished perception of aromas in food. The prevalence of OD in the general adult population is about 20% in Europe and the United States [1]. This condition is potentially dangerous because those affected are unable to detect fire, spoiled food, hazardous chemicals, and leaks of odorized natural gas [2,3]. OD may give rise to health consequences, including depression, anxiety, and social isolation. It affects quality of life by altering food enjoyment and the amount of food ingested [4].

Sniffing an odorant induces a nasal airflow, bringing odorants toward the olfactory epithelium located at the roof of the nose [2]. The olfactory epithelium (OE) contains millions of olfactory sensory neurons, each expressing an olfactory receptor at the extremities of their cilia in the nasal cavity. A receptor can bind a variety of odorant molecules [5], and an odorant molecule stimulates a combination of odorant receptors, similarly to a combination of keys on a piano generating a specific sound. This induces cellular events, generating action potentials ascending centrally through axons [6,7]. Each of these axons projects through the cribriform plate and connects to specific glomeruli in the olfactory bulb, which secretes diverse molecules to attract regenerating axons [6,7,8,9,10,11].

A particularity of olfactory sensory neurons (OSNs) is their life-long, perpetual renewal [12]. The active neurogenic niche within the olfactory epithelium features a complex cellular architecture and molecular environment [13,14,15]. The olfactory epithelium is composed of olfactory neurons in different stages of maturation, sustentacular cells (detoxifying potentially dangerous chemicals, promoting cell differentiation [14]), and olfactory progenitors lying next to the lamina propria. These progenitors can be divided into two types: horizontal basal cells (HBCs), remaining mitotically quiescent unless activated by injury, and globose basal cells (GBCs), a heterogeneous population including quiescent and active progenitors [16]. Proliferation and differentiation of these progenitors are controlled by a number of factors such as the nerve growth factor (NGF), brain-derived neurotrophic factor (BDNF) [17], glial derived neurotrophic factor (GDNF) [18], transforming growth factor-beta (TGFβ) [19], and retinoic acid [20], among others. The extracellular proteins (laminin, collagen IV, and fibronectin) found in the lamina propria and nerve fiber layer seem to be important for the attachment and growth of progenitors [21,22]. During maturation, OSN extend their axons toward the olfactory bulb. They are surrounded by olfactory ensheathing cells (OECs), which act as a scaffold for the axons to grow. They express a wide range of molecules promoting olfactory axon adhesion, growth, and guidance [23].

Olfactory axons are particularly susceptible to injury, especially after head trauma, one of the most frequent causes of smell loss [24]. For instance, an occipital impact causes rapid deceleration and movement of the brain relative to the skull, severing the axons of the olfactory neurons passing the cribriform plate and thereby leading to anosmia [25,26]. Posttraumatic gliosis and scar formation in the region of the cribriform plate probably prevent axonal regrowth toward the olfactory bulb (OB) [26,27].

There is currently no cure for this debilitating and often permanent condition. To restore smell function after a traumatic brain injury, it is necessary to (1) create an implantable scaffold allowing axonal regrowth through the scar tissues on the cribriform plate and (2) incorporate growth factors and extracellular proteins into it. However, there are several limitations and knowledge gaps that prevent us from readily determining the ideal growth factors or extracellular proteins, which motivated the design of the present study.

First, many studies about in vitro regrowth of olfactory axons use fetal bovine serum [28], containing a mixture of poorly characterized growth factors or Matrigel [29], a mixture of extracellular matrix proteins (ECM) [30]. Therefore, information on the effect of a single growth factor on axonal regrowth are scarce. In addition, the complete range of our growth factors of interest have not been screened head-to-head within a single study.

Second, regarding the type of olfactory tissues used to test these conditions, many studies utilize embryonic olfactory epithelium [15] carrying high neurogenic potential compared to an adult mouse, but this model is not representative of our patient population. Free-floating murine neurospheres could be used to screen these conditions, but they are challenging to expand in vitro, and the cellular architecture of the olfactory epithelium is lacking [29,31,32].

Finally, many methods exist to measure axonal length but none of them are “high-throughput” enough to be used in our study. For example, Neuron J requires the investigators to manually trace each single axon, which is highly time-consuming.

The aim of this study is to (1) create an automated quantification tool to measure axonal length and (2) determine the ideal growth factors and extracellular proteins to enhance axonal regrowth of olfactory sensory neurons in a mouse organotypic 2D model.

## 2. Results

### 2.1. Development of a High-Throughput Automated Quantification Tool to Measure Axonal Length

To identify the growth factors and extracellular proteins that are essential for olfactory neuron axonal elongation, we developed an automated quantification tool. We developed a MATLAB framework, which operates on pictures of β-III-tubulin stained OE explants (Figure 1A). First, the script enhances the contrast of the picture using a contrast-limited adaptive histogram equalization (Figure 1B). This first step is fundamental to increase the signal of both the explant border (Figure 1B) and the axonal tubular shapes (Figure 1B). Then, the explant is automatically segmented by intensity thresholding relying on Otsu’s method and morphological filters, followed by a Euclidean distance transform computation out of the explant. Axons are enhanced before segmentation (intensity threshold) using Hessian and top-hat filters. Finally, the binary image of the outgrowth is skeletonized, and the value of the distance transform image of these corresponding pixels is reported. The computation results are reported in an Excel file, ready to be further analyzed. The script also generates a graphical output consisting of a picture of the analyzed explant, a histogram with the amount of axonal outgrowth in mm at a given distance (Figure 1C), and an angular representation of the overall axonal outgrowth (Figure 1D). This representation provides information about the direction of the neuron’s extension.

We compared our automated quantification tool with another quantification tool based on Image J 2.0 Java 1.8 (using Neuron J) [33] that requires the investigator to manually trace each axon [34] (Figure 1E,F). The automatic quantification tool significantly outperformed Image J by reducing the analysis time by 83% (Figure 1E). Importantly, the intra class correlation coefficient (ICC) of 0.96 between the manual and the computed tracing demonstrated high accuracy in measuring axons and no other structure on the coverslips (Figure 1F).

### 2.2. The Posterior Septum and Posterior Turbinate Regions Yielded the Highest Outgrowth Rate in an Organotypic Model of Olfactory Regeneration

Four regions were used as the primary donor sites for olfactory epithelium, anterior and posterior septum (AS, PS), and anterior and posterior turbinate (AT, PT) regions. For 14 days, all these primary olfactory biopsies were cultured in 2D as organotypic cultures (Figure 2A,B) and supplemented with PM. The choice of a particular anatomical region for harvesting had an impact on the observed cellular outgrowth rates (χ^2^ (3, *n* = 80) = 8.79, *p* = 0.032, Figure 2E). PS and PT were the anatomical donor sites, yielding the highest outgrowth rates of 55% and 45%, respectively (Figure 2C,E), in contrast to the lower outgrowth rates observed with AS (15%) and AT (25%) regions. No significant difference was found for attachment rates across different harvesting regions (Figure 2D). Based on the results, the PT and PS regions were selected as the primary anatomical regions to investigate the impact of ECM proteins and growth factors on OSN regrowth.

### 2.3. Collagen IV and Fibronectin Coatings Enhance Axonal Elongation

We analyzed three distinct ECMs used as coatings on coverslips for culturing OE explants harvested from the PS and PT. These explants were then supplemented with PM media and cultured for 15 days. We measured the attachment rate, the cellular outgrowth rate, the neuronal outgrowth rate, and axonal elongation (in mm). Attachment rates were not influenced by the coating. Although the presence of cellular outgrowth was higher for fibronectin (32%) when compared to collagen IV (21%), laminin (14%), and the uncoated plastic coverslip (Ctrl), (5%), this association was not significant (Figure 3A). Regarding the neuronal outgrowth rate (as defined by βIII-tubulin positive outgrowth cells), we did not observe any associations with the extracellular proteins (fibronectin 13%, laminin 6%, collagen IV 15%, ctrl 1%; Figure 3B). In contrast, the average total axonal length was significantly enhanced in the presence of fibronectin (median: 8.1 mm; IQR: 4.7–20.4) and collagen IV (median: 7.3 mmL IQR: 3.8–16.7) when compared to the control (median: 1.6 mm; IQR: 0.6–3.4) or laminin (median: 2.1; IQR: 1.1–3.3; *p* = 0.0012 and *p* = 0.0060, respectively) (Figure 3C).

Together, optimal axonal elongation was observed in the presence of fibronectin, which was selected for subsequent GF screening (Figure 4).

### 2.4. FGF2, NGF, and GDNF Enhance Neuroproliferation in Olfactory Explants

In order to assess the influence of growth factors on our organotypic explant model, we used a two-step approach. On fibronectin-coated wells, we first used FGF2 to promote progenitor’s proliferation, and then, we used factors known to promote neuronal maturation (i.e., BDNF, GDNF, NGF, RA and TGFβ). We monitored proliferation by observing the presence of growing cells from olfactory explants, which we referred to as cellular outgrowth. In the context of organotypic models, the timing of the first appearance of cellular outgrowth can vary between specimens. To establish a normalized time-point for most biopsies to exhibit cellular outgrowth, we closely monitored the proliferation rates in OE biopsies. This allowed us to identify a specific time frame in which most of the biopsies demonstrated cellular outgrowth, thus enabling us to switch from proliferation media (containing FGF2) to differentiation media. The earliest signs of cellular outgrowth among the explants were observed after three days, and this process continued up to the 12th day (Appendix A). We observed that most of the explants (73%) displayed cellular outgrowth after seven days in culture, a time point chosen to transit to maturation media.

We observed a significantly enhanced cellular outgrowth rate upon growth factor supplementation (χ^2^ (9) = 39.97; *p* = 0.0001) (Figure 3D). In particular, FGF2 from day 0 to 7 followed by GDNF from day 7 to 15 (squared adjusted residuals = 20, *p*-value = 0.00004, adjusted *p*-value cut-off = 0.0025) or FGF2 from day 0 to 7 followed by FGF2 and NGF from day 7 to 15 (squared adjusted residuals = 20, *p*-value = 0.00007, adjusted *p*-value cut-off = 0.0025) led to a higher cellular outgrowth rate. Similarly, the rate of neuronal outgrowth within these explants, assessed by observing the presence or absence of cells positive for the βIII-tubulin marker, was significantly increased in the presence of growth factors vs. control (χ^2^ (9) =23.7; *p* = 0.005). Our results indicate that FGF2 administration throughout the entire culture duration (0–15 days) led to a neuronal outgrowth rate of 12%. While NGF alone (from days 7 to 15) achieved only 7%, the combination of FGF2 and NGF during the same period yielded the highest neuronal outgrowth rate of 33% (squared adjusted residuals = 20, *p*-value = 0.000042, adjusted *p*-value cut-off = 0.0025) (Figure 3E).

In summary, our findings demonstrate that applying FGF2, followed by either GDNF or FGF2 and NGF, significantly affects the cellular outgrowth rate in olfactory explants. However, only the latter condition, involving the combination of FGF2 and NGF, resulted in the highest rate of neuronal outgrowth. This highlights the considerable potential of this specific growth factor combination for promoting neuronal regrowth.

### 2.5. FGF2 Alone or in Combination with NGF and BDNF Enhances Axonal Elongation

Using our automated quantification tool (Figure 1), we determined the impact of growth factors on axonal elongation. While most conditions in which a single growth factor was added from 7 to 15 days did not significantly improve axonal elongation, we found that FGF2 alone or in combination with BDNF and NGF significantly enhanced the total axonal length. BDNF and GDNF also improved axonal regrowth (Figure 3F).

While several combinations of growth factors were able to promote OSN axonal elongation in our setting, the condition exhibiting the highest rate of neuronal outgrowth was FGF2 and NGF.

## 3. Discussion

The present technical study allowed us to improve the culture methods of murine olfactory neurons. The anatomical donor site for harvesting primary olfactory epithelium in mice was restricted to the posterior septum and posterior turbinate regions, thereby allowing a higher yield of olfactory neuronal outgrowth in 2D cultures. Fibronectin and collagen IV scaffold also improved axonal elongation over untreated and laminin-coated surfaces. Finally, FGF2 alone or in combination with BDNF, GDNF, or NGF exhibited a similar effect on the neuronal outgrowth and axonal elongation and outperformed single growth factor supplementation (Figure 4, summary). Additionally, we introduce a customized quantification tool tailored to address the specific requirements of this research. The custom-made automatic tool substantially reduced the time needed for the quantification of axonal elongation and outperformed the available image J-based method.

This quantification tool measures βIII-tubulin positive cells and axons surrounding the olfactory explant. It delivers an automated and comprehensive measurement of the total axonal length, which correlated highly with the measures found manually. Although it does not discriminate between axons that originate from neurons in the explants and new axons from neurons that migrated out of the explant, we believe that this tool gives valuable insight into the overall dynamics of neuronal growth.

The definition of an optimal scaffold composed of both ECM proteins and soluble factors to regrow olfactory neurons is of the utmost importance to initiating a novel therapeutic path toward regenerating the olfactory system in patients suffering from traumatic brain injury. Based on our results, it seems to be of particular interest to treat olfactory explant cultures with a combination of growth factors containing FGF2. Earlier studies claimed that FGF2 may induce differentiation in murine and human olfactory epithelium cultures [35,36]; however, this effect was later attributed to an indirect effect through the enhanced stimulation of the GBCs’ proliferation [37,38]. The specific role of this growth factor is to target proliferation in both GBCs and OECs [36] and therefore optimize the overall microenvironment for olfactory neurons to proliferate and elongate.

Our study has also shed light on the remarkable ability of FGF2 to enhance axonal elongation in vitro. However, it is crucial to consider the potential confounding factors in this observed effect. The exceptional performance of FGF2 in promoting axonal elongation may be attributed to the stimulation of progenitor and ensheathing cell proliferation. The continuous proliferation of progenitors may increase the likelihood of subsequent differentiation into neurons. Meanwhile, the ongoing proliferation and migration of ensheathing cells may facilitate the migration of these young neurons toward the edges of the quantification field, increasing the chances of βIII-tubulin-positive neurons being detected far away from the analysis starting point.

Nevertheless, this observation only partially aligns with the high expectations regarding neuronal outgrowth. A combination with a growth factor that explicitly stimulates neuronal differentiation may be necessary to achieve optimal results. While FGF2 demonstrates remarkable potential as an in vitro option for boosting progenitor and ensheathing cell proliferation, additional factors that promote neuronal differentiation are crucial for achieving the desired level of neuronal outgrowth.

Also, NGF exerts an impact on ensheathing cells, which are the glial cells located within the OE; those cells express both the low-affinity NGF receptor p75^NTR^ [18] and the NGF-specific receptor tropomyosin receptor kinase A (TrkA). Thus, the effect of NGF on ensheathing cells is mediated in part through its low affinity and in part through its specific receptor. Furthermore, studies have reported that when OSNs are co-cultured with glial cells, the retrieval of NGF from this co-culture results in a significant decrease in OSN proliferation [39]. Thus, the influence of this growth factor on axonal elongation can be partly attributed to its interaction with the ensheathing cells. Both in physiological conditions and after injury, OECs maintain their shape and structure to provide an open route to the regenerating axons [40]; thus, the axonal elongation boost observed with FGF2 and NGF can be due to their specific combined interactions with those cells.

NGF’s role in promoting axonal elongation may extend beyond its influence on OECs. Specifically, this growth factor might play a specific role in the regeneration of OSNs following injury. In support of this, Miwa et al. [41] demonstrated an increase in immunoreactivity to the NGF-specific receptor, TrkA, 21 days after olfactory nerve transection, observed in both the OB nerve layer and the OE.

When combined with FGF2, NGF elicited the highest percentages of neuronal outgrowth among all tested conditions (33%), surpassing even the effects of FGF2 alone (12%). However, the axonal elongation observed (median: 10.59 mm; IQR: 5.40–22.29) was not significantly different from that induced by FGF2 alone (median: 6.6 mm; IQR: 4.46–21.61). Despite the crucial role of NGF in OE regeneration, our study revealed that the singular administration of NGF alone is insufficient to stimulate axonal elongation effectively. This could be attributed to the fact that NGF has been shown to primarily facilitate clustering rather than actual outgrowth in organized bundles of olfactory neurons, potentially limiting its ability to promote extensive axonal elongation [42].

These observations support our hypothesis that a coordinated interplay of multiple growth factors and soluble proteins orchestrates the regeneration of OSNs. This concerted action might serve to optimize the regenerative processes involved in OSNs recovery after trauma. However, additional investigations are warranted to unravel the specific molecular mechanisms implicated when these promising combinations of soluble proteins are employed.

When considering the application of these elicitor proteins to functionalize implants for olfactory neuronal regeneration in vivo, it is crucial to account for the fibrogenic effect of FGF2, meaning the role it has in recruiting fibroblasts to the site of injury. While FGF2 may exhibit optimal activity in vitro, its use in vivo may increase scarring due to its fibrogenic properties. Further investigations are needed to elucidate how FGF2 can be replaced or its fibrogenic effect minimized.

This study introduces an optimized organotypic model specifically tailored to address regeneration of murine olfactory neurons while mitigating the potential biases associated with using poorly characterized growth factors and extracellular protein cocktails in OE cultures. Thanks to this tailored in vitro model, we demonstrated that OSN regeneration can be effectively induced, controlled, and accurately quantified using our automated quantification tool. These results hold promising implications for advancing the therapeutic paradigm of olfactory regeneration toward preclinical investigations.

## 4. Materials and Methods

### 4.1. Culture Media

Basal media (BM) was a combination of DMEM-F12 media (Sigma-Aldrich, St. Louis, MO, USA, ref: 31330038) supplemented with Hepes (15 Mm) and L-Glut (15 mM), 1× N-2 (ref. Gibco 17502048, Billings, MT, USA), B-27 (ref. Gibco 17504044), and Penicillin streptomycin (100 U/mL; Thermo Fisher Scientific, Waltham, MA, USA). Proliferation media (PM) corresponds to BM supplemented with fibroblast growth factor 2 (FGF2) 40 ng/mL (ref CYT-218 Prospec, Hamada, Israel) and was used to culture olfactory explants from days 0–7. Differentiation media (DM) was a combination of BM with various growth factors, and was used from culture days 7–15.

### 4.2. Dissection Technique to Harvest Murine Olfactory Epithelium (OE)

Harvesting techniques for OE in humans [28,29,30,43,44] and rats [45] are widely available; however they are scarce for mice [36]. This called for the development of our own harvesting surgical technique. The protocol was fully accepted by the local veterinary committee (license G110). The entire procedure was performed under a laminar flow hood with the help of a dissection microscope. Six-week-old C57BL/6 mice were anaesthetized with IP injection of Ketamine 100 mg/kg and Xylazine and then sacrificed by cervical dislocation and decapitation. After skin removal, a sagittal incision of the skull at 1–2 mm distance from the suture line was performed. In a single cut, we were able to split the skull and the nose of the mouse to expose the area of interest: septum and turbinate. Olfactory mucosa presents itself as orange/brown. We then separated the septum from the ventral connection to the vomer bone and peeled off the OE on both sides. The specimen collected from septum and turbinate were placed into falcon tubes filled with ice-cold HBSS (Hanks’ Balanced Salt Solution, Gibco) in order to both remove the mucus and store them before plating. For the sake of reproducibility, we used a Mcllwain’ Tissue Chopper (240V version; graciously provided by Alexandre Dayer’s Lab, University of Geneva, Switzerland) to obtain 200 µm thick explants.

### 4.3. Coating

Coverslips (Thermofisher 12 mm diameter) were cleaned with 70% alcohol before being placed in sterile 24-well plate for cell culture (Corning, Corning, NY, USA). Coating solutions for the different matrices were prepared in DPBS (Dulbecco’s Phosphate Buffer from Sigma) using the following concentrations: laminin 50 μg per mL (Sigma-Aldrich L2020 from Human Placenta); fibronectin 50 μg × mL (Sigma-Aldrich FC010 from human plasma [25]); collagen IV (Sigma C5533, from human placenta) 1:100 dilution of 3 mg/mL solution. We placed 500 µL of this solution in each well. Coverslips should not float, and any bubble should be eliminated. We let the solution set from 4 h to overnight at 37 °C. Explants with a diameter of 2 mm were plated onto the coated wells with forceps. One drop of basal media (BM) was applied onto the explant to keep the specimen hydrated while attaching onto the coverslip. After half an hour, BM was replaced by proliferation media (PM).

### 4.4. Growth Factor Treatments

The growth factors (GFs) used in the study, with reference numbers, supplier information, and concentrations are shown in Table 1. When using combinations of growth factors (BDNF and GDNF; FGF2 and NGF; FGF2 and BDNF), the same concentrations of individual growth factors were used.

### 4.5. Growth Factors Administration Paradigm

To harmonize the cellular outgrowth frequency among explants, we monitored the biopsies; proliferation rate along the 14 days of culture: 30% of biopsies were giving visible outgrowth already after day 5, more than a half of explants after day 7 (63%), 74% after day 9, and 81% after day 12 (Appendix A, x axis: number of explants observed giving cells outgrowth; y axis: day of culture). Based on these results, we decided to perform the switch between the proliferating media (PM) and the differentiation media (DM) after day 7 of culture in order to exploit to the maximum the proliferation power of progenitors within the biopsies before driving them into differentiation.

### 4.6. Immunostaining

After removal of culture media, the coverslips were washed with 500 µL PBS and then fixed with 300 µL Formol 4% for 15 min. Specimens were permeabilized (0.2% Triton ×100 in PBS) for 20 min at room temperature and treated with a blocking solution (1% Bovine Serum Albumine, 0.1% Triton ×100 in PBS) for another 20 min. Primary antibodies (dilutions and reference in Table 2) were diluted in blocking solution and incubated overnight at 4 °C. The following day, explants were rinsed three times for 10 min with PBS and incubated for 2 h at room temperature with the secondary antibodies (dilutions and references in Table 3) in blocking buffer. The stained samples were washed three times with PBS and mounted on glass slides with Fluoroshield containing DAPI (Sigma-Aldrich).

### 4.7. Imaging Process

Stained samples were subjected to automated imaging using a Zeiss Axioscan. Z1 slide scanner (Zeiss, Oberkochen, Germany) equipped with a Hamamatsu Orca Flash 4 camera and a Zeiss 20× objective lens (Plan-Apochromat 20×/0.8 M27) with a resolution of 0.325 microns/pixel. Slides containing three specimens each were loaded onto the scanner to enable simultaneous image acquisition, leveraging its high capacity to accommodate up to 100 slides. Before initiating the acquisition, the region of interest was determined for each specimen. The acquisition process adhered to the predefined parameters outlined in the Scanning profile, encompassing the acquisition of 9 sections per specimen (z-stack) with intervals of 2.50 µm. The Extended Deep Focus technique used the “Variance” method to capture detailed images. The primary focus during image acquisition was directed toward βIII-tubuline-positive cells, with the secondary antibody conjugated to Alexa Fluor 555 Donkey (Life Technologies—Thermo Fisher Scientific, Geneva, Switzerland). To achieve this, a filter with excitation at 540 nm and emission at 561 nm was used. The light source intensity was 58.32%, and the exposure time was fixed at 100 ms (Figure 1A). Images presenting acquisition defects that could bias the computation were not included in the study.

### 4.8. Data Collection and Statistical Analysis

As outcome measures, we evaluated the attachment rate, cellular outgrowth rate, neuronal outgrowth rate, and axonal elongation (in mm). Cellular outgrowth refers to the presence or absence within the culture of a general cellular outgrowth and can be evaluated using a simple bright field microscope (representative picture in Appendix A). Neuronal outgrowth was defined by the presence versus absence of βIII-tubulin staining around the explants. While all the explant outgrowth was positive for DAPI staining (a nuclear marker), not all of them were positive for βIII-tubulin staining. βIII-tubulin is distinctively confined to neurons during the initial stages of neuronal differentiation.

Statistical analysis was performed using GraphPad Prism 9.4.1.681 and IBM SPSS Statistics 25. We used the Pearson chi-square test of independence to find variables (biopsy anatomical regions, ECM, and growth factors) associated with explant attachment and cellular or neuronal outgrowth. For post hoc analysis of Pearson’s chi-square, we calculated squared adjusted residuals (or z-square) and transformed them into *p*-values using a formula integrated into SPSS. The significance level of 0.05 was adjusted by dividing it by the number of associations, and transformed *p*-values were compared to this cut-off to reject the null hypothesis [47].

Since the axonal elongation data were not parametrically distributed, we performed a logarithmic transformation on the axonal elongation and then analyzed it [48,49] with one-way ANOVA multiple comparison to check if there was a significant difference among the standard deviations of every condition. As previously outlined (Bland and Altman, 1996 [48]), the acquired values were reverted to their original form after the analysis to facilitate graphical representation. The graphical data in Figure 3C, F showcase the axonal elongation in millimeters while the statistical analysis was performed on the transformed data.

## Figures and Tables

**Figure 1 ijms-24-12863-f001:**
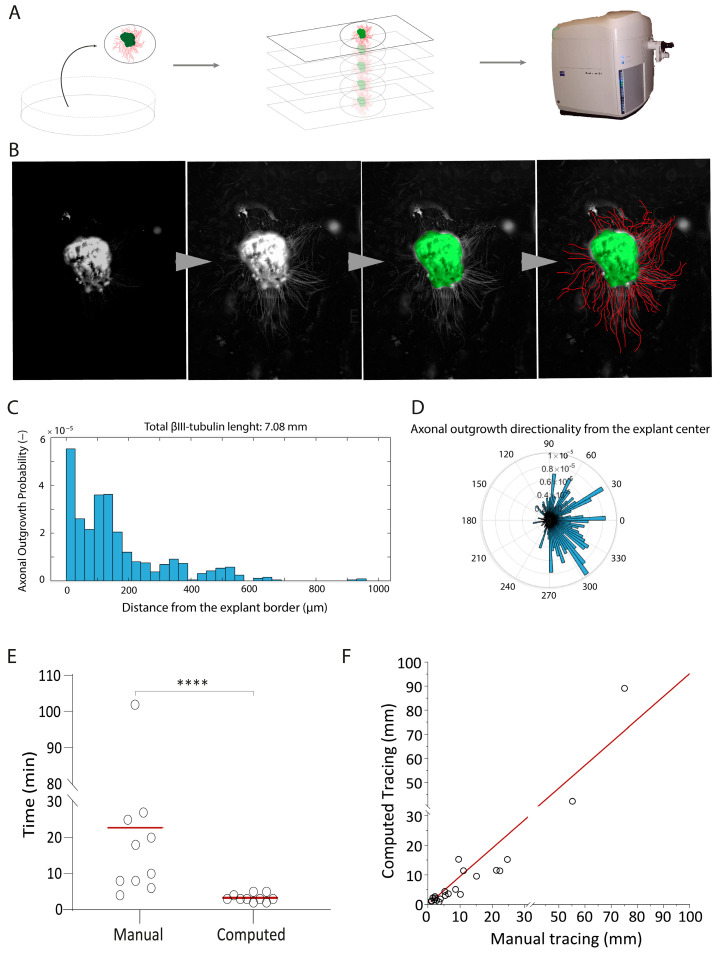
The development of an automated axonal elongation quantification tool. (**A**) Samples imaging paradigm: the method involves placing fixed and stained explants onto slides, which are imaged using the Axioscan. Z1 platform. The primary focus of image acquisition is directed toward Tuj-positive cells. (**B**) Steps involving MATLAB script processing: the progression from left to right encompasses artifact removal and fluorescence enhancement, defining the borders of explants, and tracing axons within the specimens. (**C**) Example of computational output yielded by the employed tool. This output consists of a histogram graph that portrays the detected fluorescence intensity (y axis) in relation to the distance (x axis). Additionally, the top portion of the graph displays the total length of axonal elongation, measured in millimeters. (**D**) Histogram showing directional preference of axonal growth from the explant’s border. This outcome measure was not used in the present study, but shows the potential of the measurement tool for other groups or future experiments. (**E**) Dot plot to compare the manual tracing of axons versus the application of the automated tool. The statistical analysis using a *t*-test yields a significant mean **** = *p*-value ≤ 0.0001. (**F**) Intra class correlation (ICC) analysis, which yields a high ICC value of 0.96. The accompanying 95% confidence interval (CI) is reported as ±0.014, underscoring the reliability of the measurements.

**Figure 2 ijms-24-12863-f002:**
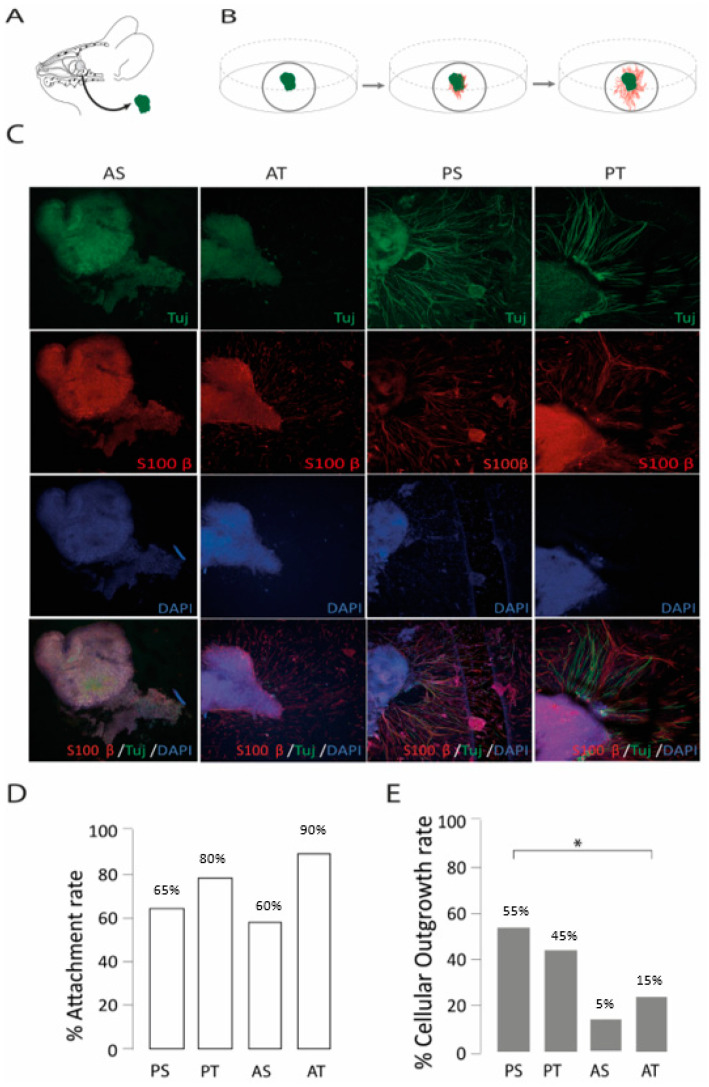
The development of a mouse organotypic 2D model of olfactory regeneration. (**A**) Biopsies were harvested from the intranasal mucosa and (**B**) cultured on glass coverslips. (**C**) Confocal images of olfactory epithelium biopsies harvested form posterior septum (PS), posterior turbinate (PT), anterior septum (AS), and anterior turbinate (AT), where outgrowth was mainly visible from samples of PS and PT. βIII-tubulin (Tuj in green) was used as a neuronal marker, S100β (in red) as an ensheating cell marker, and DAPI (blue) to stain for cell nuclei. (**D**,**E**) Bar charts depicting the attachment (*n* = 20 per condition) and cellular outgrowth rates (*n* = 20 per condition) by anatomical regions. * = *p* ≤ 0.05.

**Figure 3 ijms-24-12863-f003:**
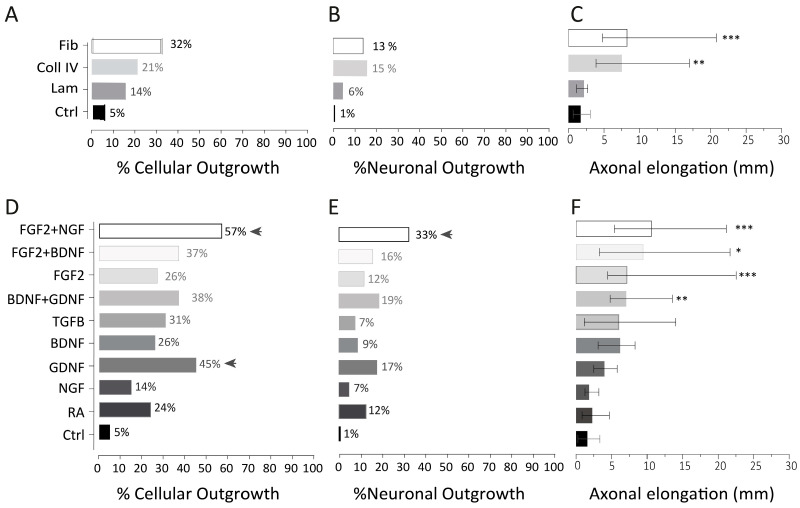
Effect of individual extracellular matrix (ECM) proteins and growth factors (GFs) on olfactory neuron regeneration. (**A**–**C**) The first row shows the effect of ECMs on cellular outgrowth (*n* = 84 per condition), neuronal outgrowth (*n* = 84 per condition), and axonal elongation (*n* = 13 per condition; median and interquartile). Cellular outgrowth refers to the presence or absence within the culture of a general cellular outgrowth and can be evaluated using a simple bright field microscope (representative picture in Appendix A). Neuronal outgrowth was defined by the presence versus absence of βIII-tubulin staining around the explants. While all the explant outgrowth was positive for DAPI staining (a nuclear marker), not all of them were positive for βIII-tubulin staining. βIII-tubulin is distinctively confined to neurons during initial stages of neuronal differentiation. (**D**–**F**) The second row shows the effect of GF on cellular outgrowth (*n* = 42), neuronal outgrowth (*n* = 42), and axonal elongation (*n* = 12; FGF2 + BDNF *n* = 6, NGF *n* = 9, TGFB *n* = 7; median, interquartile). Since the administration of GF from day 0 to 7 is consistent across all conditions (fibroblast growth factor 2), except for control group in which no GF was administered, the growth factors represented in the y-axis refer to those administered from day 7 to 15. Post hoc analysis of Pearson’s chi-square adjusted residuals highlighted significant conditions, marked by arrow head. * = *p* ≤ 0.05; ** = *p* ≤ 0.01; *** = *p* ≤ 0.001; fib = fibronectin; Coll IV = collagen IV; Lam = laminin; Ctrl = uncoated plastic coverslip.

**Figure 4 ijms-24-12863-f004:**
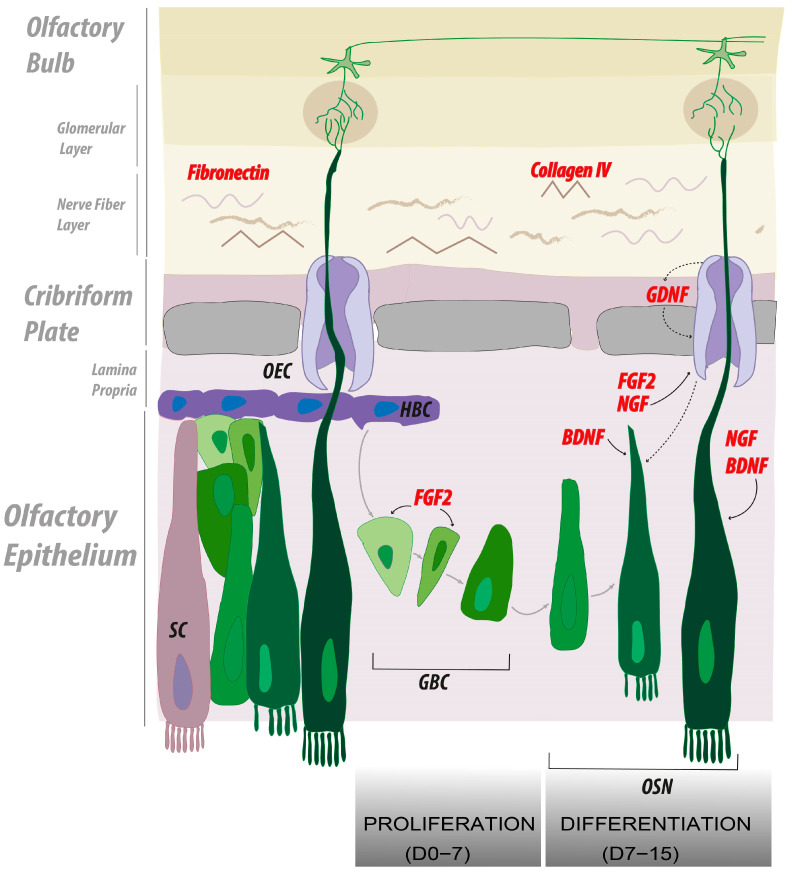
Summary of the main findings representing the cellular and molecular interplay leading to OSN regeneration. In our study, we screened both soluble extra cellular matrices (ECMs) and growth factors (GFs) to determine the best in vitro culture conditions to facilitate olfactory neuron axonal regrowth. Most significant candidates are depicted in red. Our model highlights the role of the ECM components, fibronectin and collagen IV, as support for olfactory neuron regrowth, further stimulated by the soluble factors, nerve growth factor (NGF) and fibroblast growth factor 2 (FGF2). In vivo, the proliferation of globose basal cells is enhanced by the action of FGF2. We emulated this first phase by administrating FGF2 from day 0 to day 7 of culture. In a second phase, we directed olfactory neurons differentiation using NGF, brain-derived neurotrophic factor (BDNF), and glia-derived neurotrophic factor (GDNF). We recapitulated this phase in vitro from day 7 to 15 of culture.

**Table 1 ijms-24-12863-t001:** Growth factors list.

Name	Reference Number	Supplier	Concentration
Fibroblast growth factor 2 (FGF_2_)	CYT-218	Prospec	40 ng/mL [36]
Retinoic acid (RA)	R2625-50MG	Sigma-Aldrich	10 ng/mL [20]
Nerve growth factor (NGF)	N6009-4X25UG	Sigma-Aldrich	50 ng/mL
Brain-derived neurotrophic factor (BDNF)	B3795-10UG	Sigma-Aldrich	50 ng/mL [17]
Glial cell-derived neurotrophic factor (GDNF)	450-10	Perotech (Chapel Hill, NC, USA)	20 ng/mL [23]
Transforming growth factor β(TGFβ)	T2815-2UG	Sigma-Aldrich	10 ng/mL [46]

**Table 2 ijms-24-12863-t002:** Primary antibodies list.

Antibody Name	Reference Number	Supplier	Working Dilution
S-100 beta	S2532-100UL	Sigma-Aldrich (Merck)	1/800
Tubuline beta 3 (TUJ1)	802001	Biolegend (San Diego, CA, USA)	1/1000

**Table 3 ijms-24-12863-t003:** Secondary antibodies list.

Antibody Name	Reference Number	Supplier	Working Dilution
Alexa fluor 488 Donkey	A32766	Life Technologies (Carlsbad, CA, USA)	1/1000
Alexa fluor 555 Donkey	A31572	Life Technologies	1/1000

## Data Availability

Not applicable.

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
