# Peer review of "Axonal Regrowth of Olfactory Sensory Neurons In Vitro"

_ijms, 2023, doi:10.3390/ijms241612863_

Round 1
Reviewer 1 Report
In the manuscript titled, "Axonal regrowth of olfactory sensory neurons in vitro", the authors test different conditions for the optimal growth of olfactory sensory neurons in vitro, working toward the ultimate goal of designing therapeutic strategies for the regeneration of olfactory neurons after traumatic injury. While the study lacks novelty, the experimental and results were presented adequately. I have the following comments and suggestions for improvement:
---- More descriptions and explanations are needed in figure legends in general.
---- In Figure 1D, which shows an example output of the directionality of axon outgrowth for a single explant, please explain if the direction relative to any specific topological feature of the explant, external cues or randomly assigned. The information gained from this analysis while valuable, is not used in subsequent experiments.
---- In Figure 1F, broken axes are used for both x and y axes, but the breaks are not in the same position for both and the correlation is shown as an unbroken line. In my opinion this is misrepresenting the data.
---- It is not clear how statistics for determining significant differences were performed for figures 2D-E, 3A-B, 3D-E, as these are percentage values with no error bars.
---- Table 1 contains data similar to that shown in Figure 3, albeit different condition. It would be nice if these were shown in plots similar to Figure 3 for visual comparison instead of a table.
Author Response
For the overall point-by-point response, please see the attachment.

Reviewer 2 Report
In this paper, the authors have proposed an automated quantification tool to measure axonal length and tested several growth factors and soluble proteins for regeneration of olfactory sensory neurons.
- Affiliations: please provide the complete address information including zip code.
- Abbreviations should be defined the first time:
Abstract, pg. 1: FGF2, BDNF, GDNF, NGF, TGF-β
Results, pg. 5: ECM proteins and OSN regrowth
Discussion: pg. 11: OB nerve layer
- Please replace in-vito with in vitro.
- Correct Collagen Iv from pg 7.
- 4.6 Immunostaining: Please replace ul with µl.

Author Response

(The authors gave the same response as above.)

Reviewer 3 Report
The authors describe a nice organotypic cell culture model for olfactory epithelium biopsies and respective cellular and neurite outgrowth assay evaluation. For this they set-up an automatic quantification tool that will alos be of interest for readers working with similar cell cultre models, e.g. dorsal root ganglion neurite outgrowth assays.
There is just two things that should be carefully addressed during revision:
1) it is not sufficiently clear, how the authors differentiated between cellular and neuronal outgrowth? Was there a specific staining used for visualising and quantifying proliferating neurons in contrast to all proliferating cell types and all cell types that migrated out of explants?
2) it is also not sufficiently clear if there is a pure effect of growth factor supplemenation of axonal elongation with regard to single neurons (e.g. each neurons extends longer axons) or if with neuronal migration and successive neurite extension the distance between outer border axons and center of the explant is extended, while the single neuron that has migrated out of the explant has grown axons with lengths similar to neurons inside the explant?
Minor:
1) Figure legends should be checked for minor language issues. Further they should stand alone and explain all abbreviations used in the figures.
2) FGF2 should be written with capital "2" following the official nomenclature.
Author Response

(The authors gave the same response as above.)
